# Biologics in Severe Eosinophilic Asthma: Three-Year Follow-Up in a SANI Single Center

**DOI:** 10.3390/biomedicines10020200

**Published:** 2022-01-18

**Authors:** Paolo Solidoro, Stefania Nicola, Irene Ridolfi, Giorgio Walter Canonica, Francesco Blasi, Pierluigi Paggiaro, Enrico Heffler, Diego Bagnasco, Filippo Patrucco, Fulvia Ribolla, Caterina Bucca, Giovanni Rolla, Carlo Albera, Luisa Brussino

**Affiliations:** 1S.C. Pneumologia U, Azienda Ospedaliero-Universitaria Città della Salute e della Scienza, Corso Bramante, 88, 10126 Turin, Italy; psolidoro@cittadellasalute.to.it (P.S.); filippo.patrucco@unito.it (F.P.); fulvia.ribolla@unito.it (F.R.); carlo.albera@unito.it (C.A.); 2Department of Medical Sciences, University of Turin, Corso Achille Mario Dogliotti, 14, 10126 Turin, Italy; irene.ridolfi@unito.it (I.R.); caterina.bucca@unito.it (C.B.); giovanni.rolla@unito.it (G.R.); luisa.brussino@unito.it (L.B.); 3Personalized Medicine, Asthma and Allergy-IRCCS Humanitas Research Hospital, Via Alessandro Manzoni, 56, 20089 Rozzano, Italy; giorgio_walter.canonica@hunimed.eu (G.W.C.); enrico.heffler@hunimed.eu (E.H.); 4Department of Biomedical Sciences, Humanitas University, 20090 Pieve Emanuele, Italy; 5Fondazione IRCCS Ca’ Granda Ospedale Maggiore Policlinico, Respiratory Unit and Cystic Fibrosis Adult Center, 20122 Milan, Italy; francesco.blasi@unimi.it; 6Department of Pathophysiology and Transplantation, University of Milan, 20122 Milan, Italy; 7Department of Surgery, Medicine, Molecular Biology and Critical Care, University of Pisa, 56124 Pisa, Italy; pierluigi.paggiaro@unipi.it; 8Allergy and Respiratory Diseases, IRCCS Policlinico San Martino, University of Genoa, 16132 Genoa, Italy; dott.diegobagnasco@gmail.com; 9S.S.D.D.U. Allergologia e Immunologia Clinica, AO Ordine Mauriziano Umberto I, 10128 Turin, Italy

**Keywords:** severe eosinophilic asthma, biologics, T2-high asthma, SANI registry, IL-5, IL-4, IL-13, omalizumab, mepolizumab, benralizumab, dupilumab, ACT, F_E_NO, lung function tests, real-life setting

## Abstract

Introduction: Biologic drugs have dramatically improved severe eosinophilic asthma (SEA) outcomes. Our aim was to evaluate the long-term efficacy of biological therapy in SEA in a real-life setting and to identify the predictors for switching to another biological drug in patients with poor asthma control. The outcomes for efficacy were decreased annual exacerbations (AE) and improved asthma control test (ACT). Methods: In 90 SEA patients being treated with a biological drug, clinical examination, ACT, blood eosinophils count and spirometry were assessed before (T0) and after 6 (T1), 12 (T2), 24 (T3) and 36 (T4) months from the start of biological therapy. Patients were considered responders (R) or non-responders (NR) to biologics depending on whether or not they had less than two AE and a 20% increase in the ACT after 12 months of treatment. Results: 75% of the patients were R, 25% NR. In R patients, biological therapy add-on was followed by significant improvement in AE and ACT throughout the whole follow-up period. The percentage of patients on oral corticosteroids (OCS) dropped from 40% to 12%. By contrast, the NR patients were shifted to another biological drug after 12 months of therapy, as they still had high AE and nearly unchanged ACT; 40% of them still needed OCS treatment. The predictors of switching to another biological drug were three or more AE, ACT below 17, nasal polyposis and former smoking (*p* < 0.05). In NR, the shift to another biological drug was followed by a significant decrease in AE and an increase in the ACT. Discussion: This real-life study confirms the long-term efficacy of biologics in most SEA patients and indicates that even in non-responders to a first biological drug, it is worth trying a second one. It is hoped that the availability of additional biologics with different targets will help improve the personalization of SEA therapy.

## 1. Introduction

Asthma is the most common chronic inflammatory airway disease, affecting over 300 million people worldwide [1]. The mild or moderate form of the disease affects over 90% of asthmatics, while severe asthma (SA) affects about 5–10% of patients [2]. According to the 2021 Global Initiative for Asthma (GINA) [3] definition, severe asthma means “asthma that is uncontrolled despite adherence with maximal optimized therapy and treatment of contributory factors, or that worsen when high dose treatment is decreased”. A recent Italian study [4] showed that up to two-thirds of patients with SA usually take oral corticosteroids (OCS) daily for improving asthma control or at least during exacerbations [5,6]. Poor asthma control and the consequent OCS overuse represent a public health burden in terms of economic impact and patients’ quality of life. In the last 20 years, many monoclonal antibodies have become available as add-on treatments for patients with severe asthma (steps 4 and 5 according to the GINA guidelines) (1), and their introduction yielded a progressive reduction not only in asthma exacerbations and hospitalizations but also in OCS consumption. Four biologics that act on different targets, such as cytokines (Interleukin (IL)-5, IL-5R, IL4-Rα) and immunoglobulin E (IgE), are available at present in Italy for the treatment of severe eosinophilic asthma (SEA). The specific targets of approved biologics make these drugs effective only in type 2 asthma, hence the need to properly classify severe asthma patients according to their specific inflammatory endotype [1,7]. Several clinical trials demonstrated the efficacy of biologics in patients with severe eosinophilic asthma [8,9,10,11], in terms of reduction in exacerbations and hospitalization rates, improvement in quality of life, and lung function tests. Unfortunately, such studies are often limited to 48- or 52-week follow-up, such that the long-term efficacy of these drugs has yet to be definitively demonstrated.

The purpose of our study was to retrospectively evaluate the efficacy of biologics in a real-life setting, in terms of improvement of the annual exacerbations rate and asthma control assessed by the Asthma Control Test (ACT), and to identify the predictors for switching from one biological drug to another in patients who continued to have poor asthma control. To this end, all of the adult immunocompetent SEA patients included in the Severe Asthma Network Italy (SANI) center of Turin were enrolled.

## 2. Materials and Methods

### 2.1. Study Design and Participants

The study was a single-center Italian retrospective study of 90 adult SEA patients, included in the SANI center in Turin, Italy, who were receiving a biological treatment. The enrollment was performed in the period between January 2017 and May 2021.

All of the enrolled patients provided their written informed consent. The study was approved by the ethics committee (“Area Vasta Nord-Ovest Toscana”, protocol number 73714, study number 1245/2016), and it was conducted according to the Helsinki Declaration.

Clinical examination, history, annual number of asthma exacerbations (AE) and hospitalizations, ongoing treatment, ACT questionnaire [12], lung function tests, Fraction Exhaled Nitric Oxide (F_E_NO) measurements, and blood eosinophil counts were assessed in all patients before (T0), and after 6 months (T1), 12 months (T2), 24 months (T3) and 36 months (T4) from the start of biological therapy.

At each follow-up scheduled visit, the importance of careful compliance was stressed, and the correct inhalation technique was verified.

After 12 months of biologic treatment, the patients were classified as responders (R) or non-responders (NR) depending on whether or not they had an annual exacerbations rate below 2 and an increase in ACT of at least 20%. NR patients were then switched to a different biologic drug.

### 2.2. Lung Function Tests

Lung function tests were performed using a Baires System (Biomedin, Padua, Italy) according to the American Thoracic Society guidelines [13,14]. At least three different flow–volume curves with vital capacity (VC) within 5% were obtained. The values of prebronchodilator VC, Forced Expiratory Volume in the 1st second (FEV1) and FEV1/VC% ratio were recorded. Data were expressed as percentages of predicted values.

Bronchodilator response was expressed according to the GINA guidelines [1], as an increase of 200 mL or more and 12% or more from baseline.

### 2.3. F_E_NO

F_E_NO was measured according to the ERS/ATS guidelines [13] using a NO electrochemical analyzer (Hypair, Medisoft, Sorinnes, Belgium).

### 2.4. Asthma Exacerbations and Hospitalization Rate

Asthma exacerbations were defined according to the ATS/ERS joint statement [15] based on unscheduled physician visits for acute or subacute worsening of respiratory symptoms, associated with airflow obstruction, requiring a change or increased dose of medications, need for oral corticosteroid or antibiotics, and/or hospitalization. AE were expressed as the number of exacerbations between two consecutive follow-up scheduled visits. The hospitalization rate was defined as the number of asthma-related hospital admissions between two consecutive follow-up scheduled visits.

### 2.5. Statistical Analysis

Statistical analysis was performed using the statistical package IBM SPSS Statistics for Windows, version 26 (IBM Corp., Armonk, NY, USA). Normality distribution of data was first tested using the Kolmogorov–Smirnov normality test, then a descriptive analysis of the variables was performed.

Baseline characteristics were evaluated in the whole cohort and expressed as mean (±standard deviation, SD), unless specified, for continuous variables and as absolute and relative frequencies for categorical variables. Baseline characteristics of R were compared to those of NR using the Student *T*-test.

The number of exacerbations and hospitalizations at 6 months (T1) was annualized by multiplying it by two.

The effects of biologics on asthma control were evaluated using a mixed model for repeated measures (MMRM) analysis, having as dependent variables AE and ACT.

Interaction p-values at a 10% threshold level (*p* < 0.10) were calculated to assess the effect of biologics across the subgroups [16].

The predictors of switching biologics were assessed using multivariate logistic regression analysis, with non-response as the dependent variable.

The predictors were expressed as odds ratio (OR) with 95% confidence intervals (CI). The *p*-values below 0.05 were considered statistically significant.

## 3. Results

The 90 patients enrolled in the study included 59 women and 31 men, with a mean age of 59.5 years (range 18–80). As shown in Figure 1, out of the 90 patients, 73 completed 12 months, 53 completed 24 months, and 39 completed 36 months of biological treatment. Demographic data and comorbidities of the enrolled patients are listed in Table 1.

Most patients were normal weight, and 11 (12%) were obese. The most common comorbidities were rhinitis found in two-thirds of the patients, nasal polyps in nearly half, and gastroesophageal reflux disease in one-third.

All patients were taking inhaled corticosteroid/long-acting β-agonist (ICS/LABA) combination, which was associated with a long-acting muscarinic antagonists (LAMA) in 30 (33%) and with a leukotriene receptor antagonists (LTRA) in 36 (40%); 37 (41%) subjects were also receiving OCS therapy.

The choice of the biological drug depended not only on the indications for prescribing a specific monoclonal antibody but also on its reimbursement by the national health system. As shown in Figure 1, 40 patients (44.4%), received omalizumab (available in Italy since 2006), 29 (32.2%) received mepolizumab (available since 2017), 14 (15.6%) received benralizumab (available since 2019) and 7 (7.8%) received dupilumab (available since 2020).

The evaluation of the effectiveness of biological therapy, performed after the first year of therapy (T2), showed that most patients (55 out of 73 = 75%) were responders, while 18 patients (25%) were NR to the first biological treatment and were switched to another one.

### 3.1. Comparison between Responders and Non-Responders

Table 1 lists baseline data before biologics add-ons in the overall patient cohort and the comparisons between R and NR patients. No significant difference was found between R and NR in anthropometric data and distribution of comorbidities, apart from a greater prevalence of former smokers in the NR group.

Table 2 lists the trends throughout the follow-up period in the outcome variables (AE and ACT), FEV1 (% of predicted value), FEV1/VC% ratio, and F_E_NO, and in the prevalence of patients on chronic OCS in the overall cohort and in the R and NR groups. As there were very few hospitalizations, their number was included in the exacerbations rate.

In baseline conditions, the NR group had a significantly higher AE rate and lower ACT and FEV1. These differences persisted and became even more marked 6 (T1) and 12 months (T2) after the start of biological treatment, such that at this time, NR were switched to another biological drug. The change in treatment was followed by a significant improvement in AE, ACT and FEV1, which persisted up to the 36-month follow-up visit. Actually, after 2 years of treatment with the new biological drug, the only difference between NR and R was a slightly lower ACT. However, in the NR group, the percentage of patients who needed OCS remained the same and was consistently higher than that found in R patients throughout the follow-up.

### 3.2. Trends in the Analyzed Variables over Time in Responders and Non-Responders

The changes from baseline in the outcome variables (AE and ACT) at each follow-up visit in R and NR patients, evaluated by MMRM analysis, are reported in Table 3. In R patients, the improvements in AE and ACT were always significant, starting from the 6th up to the 36th month of biological drug treatment. By contrast, in NR patients, both AE and ACT remained unchanged after 1 year of treatment with the first biological drug, but improved significantly after the switch to another biological drug, and that improvement persisted up to the last follow-up visit.

The changes from baseline in predicted FEV1% and in F_E_NO throughout the follow-up are reported in Appendix A. FEV1 showed a significant improvement at the 24-month visit in both R and NR groups. In R patients, F_E_NO was significantly decreased after biologics add-ons from the T1 up to the T4 visit, while in NR patients it remained unchanged throughout the follow-up.

### 3.3. OCS Therapy

Before biologics add-ons, nearly half of the patients (46%) depended on maintenance oral corticosteroids, with a slightly higher prevalence in NR than in R (59% vs. 41%). However, in Rs the prevalence dropped to 18% after 12 months and further halved to 12% after 36 months, while in NRs it remained around 40% throughout the study.

### 3.4. Predictive Factors for Switching Biologics

Multivariate analysis (Table 4) showed that predictive factors for switching biologics were having three or more AE/year, an ACT score lower than 17, nasal polyps, and status as a former smoker (*p* < 0.05).

## 4. Discussion

This is the first report, in a real-life setting, on the efficacy of 36 months of treatment with biological drugs in patients with severe eosinophilic asthma. The results of the study indicate that in most SEA patients biologic therapy add-on was followed by a significant improvement in asthma exacerbations rate and ACT score.

The characteristics of our sample were those expected in patients with SEA, i.e., high prevalence of women (66%) [17], atopy (71%) [18], rhinitis (73%), and nasal polyps (53%) [19].

We set as primary outcomes annual exacerbations rate and ACT score. Asthma exacerbations are still a major health risk, associated with substantial healthcare costs, poor quality of life and lung function impairment [20,21]. Moreover, asthma exacerbations represent one of the main causes of increased OCS dose and admission to intensive care, leading to increased asthma-related costs, and progressive respiratory impairment. Therefore, it is essential that the effectiveness of a treatment be evaluated on its ability to reduce exacerbations. Several randomized controlled trials, as well as real-life studies using AE as an indicator of treatment efficacy, demonstrate the efficacy of biologics in reducing AE [9,22,23]. In addition to the AE rate, we also used the ACT score as an outcome variable, because it provides a subjective evaluation of asthma control. The ACT is a self-administered tool that is easy for patients to complete, and it is now recommended not only as a measure of severe asthma monitoring [24] and management [25] but also as a guide for switching to alternative treatments [26] when the disease is poorly controlled.

Seventy-five percent of our SEA patients responded to the first biologic treatment, showing a significant decrease in AE and an increase in the ACT score. The improvement was consistent, rapid, manifesting itself already after the first 6 months of therapy, and sustained, persisting up to 36 months of therapy. The improvement involved also airway obstruction and inflammation, as shown by the significant increase in FEV1 and decrease in F_E_NO throughout the study. These findings support the efficacy of biologics in severe eosinophilic asthma in real-life, in line with data of other studies [27,28,29].

However, a quarter of the patients received no benefit from 1 year of biologic therapy, in agreement with the results of recent studies [30,31]. Still, the anthropometric characteristics and prevalence of comorbidities in these NR patients were similar to those in the responders [32], apart from a higher prevalence of former smokers (23% vs. 4%). Indeed, in the first 6 months of therapy, a slight although not significant improvement in AE and ACT score was observed also in NR patients, but they showed no further improvement after 1 year of treatment. This suggests that the initial improvement was merely due to the increase in therapy compliance resulting from closer patient monitoring [33], rather than being an effect of the biological drug. Interestingly, the consequent switch to another biological drug offered to NR patients produced a significant and persistent decrease in AE already apparent after 1 year of therapy. Interestingly, in these patients at the 36th-month visit (2 years after the therapeutic change), the number of exacerbations was similar to that recorded in R patients. The mean ACT score was also significantly increased from baseline in NR patients, but it still remained lower than in R patients up to the end of the study.

Oral corticosteroid therapy deserves special consideration. In baseline conditions, nearly half of the patients depended on OCS, but after biologics add-on, the OCS could be withdrawn in most R patients, confirming the OCS-sparing effect of these drugs. Actually, in these patients, the percentage of those taking OCS dropped from 41% to 18% after 12 months and further to 12% after 36 months. The steroid-sparing effect is very important when considering the costs of OCS side-effects [34] and the increase in mortality that was recently reported in patients with severe asthma who overused OCS [35]. Unfortunately, the prevalence of NR subjects who continued to require OCS remained unchanged (around 40%) until the end of the study. The failure in OCS withdrawal and in achieving satisfactory disease control suggests that some NR patients had a suboptimal response also to the second monoclonal antibody treatment.

According to the results of the multivariate analysis, the independent predictors for poor response to biological drugs were high AE number (3 or more/year), low ACT score (below 17), nasal polyps, and past smoking. These findings are not surprising, as high AE and a low ACT score represent important red flags for more severe disease [36], and nasal polyposis is a well-known factor for asthma severity [19,37].

It should be emphasized that in most patients the choice, at the first prescription, was limited to two biological drugs. We do not know if the availability of other biologics, such as dupilumab for patients with nasal polyposis, could have improved asthma control even in NR.

We acknowledge that our study has some limitations. Actually, it has all the limitations relative to a real-life study, in which biologic prescription is non-randomized, but rather imposed by drug availability, making impossible a head-to-head comparison among biologics. Moreover, due to the staggered recruitment, only a part of the initial cohort completed the 36-month follow-up. Nevertheless, the long-term follow-up allowed us to identify the clinical variables associated with treatment failure and the consequent need for switching biologics. We believe that the problems we encountered in prescribing biologics are shared by all centers involved in the treatment of severe asthma.

On the other hand, a real-life setting is the most reliable way for objective evaluation of biological treatment efficacy.

## 5. Conclusions

Our real-life study indicates that most SEA patients gain long-term benefits from biological therapy, including a significant decrease in exacerbations, improvement in the ACT score, and a decrease in OCS use. Unfortunately, a minority of patients show no benefit from treatment. The characteristics associated with poor biologics response are high AE, low ACT score, nasal polyposis, and persistent OCS dependency. Some of these patients may benefit from the switch to another biological drug, but the type and timing of treatment change should be a focus for future investigation when a greater variety of biological drugs are available.

## Figures and Tables

**Figure 1 biomedicines-10-00200-f001:**
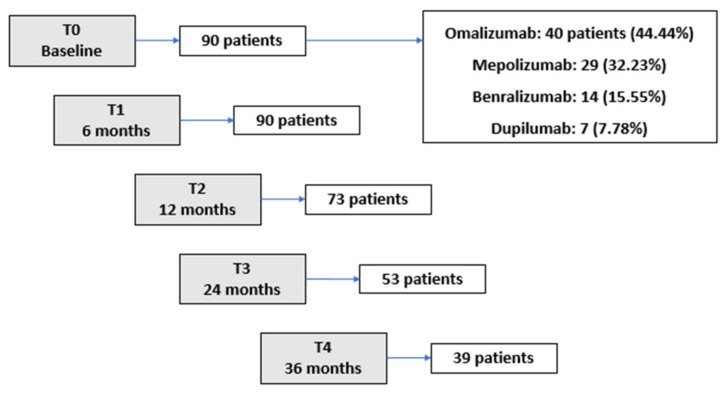
Number of patients who completed the follow-up period.

**Table 1 biomedicines-10-00200-t001:** Demographic data and comorbidities of the enrolled sample.

	Overall Patients	Responders	Non-Responders	*p*
Patients enrolled, n	90	68	22	n.s.
Age, yrs (range)	59.5 (18–80)	60.7 (33–80)	58.3 (39–81)	n.s.
Age at onset of asthma, yrs (range)	33.9 (3–70)	34.9 (3–70)	31.1 (7–55)	n.s.
Females, n (%)	59 (65.6)	46 (67.6)	13 (59.1)	n.s.
BMI	24.7 ± 4.1	24.6 ± 4.0	24.99 ± 4.4	n.s.
Patients with atopy, n (%)	64 (71.1)	49 (72.1)	15 (68.2)	n.s.
Smoking habits				
Non-smokers, n (%)	76 (84.4)	60 (88.2)	16 (72.7)	n.s.
Smokers, n (%)	6 (6.7)	5 (7.3)	1 (4.5)	n.s.
Former smokers, n (%)	8 (8.9)	3 (4.4)	5 (22.7)	0.012
OCS				
Daily dose (mg ± SD)	16.6 ± 10.1	17.0 ± 10.9	15.5 ± 8.7	n.s.
Patients treated, n (%)	41 (45.5)	28 (41.2)	13 (59.1)	n.s.
Rhinitis, n (%)	66 (73.3)	49 (72.1)	17 (77.3)	n.s.
Nasal Polyps, n (%)	48 (53.3)	37 (54.4)	11 (50.0)	n.s.
Gastroesophageal reflux disease, n (%)	25 (27.8)	17 (25.0)	8 (36.4)	n.s.
Atopic dermatitis, n (%)	5 (5.6)	4 (5.9)	1 (4.5)	n.s.
Arterial hypertension, n (%)	16 (17.8)	14 (20.6)	2 (9.1)	n.s.
Obesity, n (%)	11 (12.2)	9 (13.2)	2 (9.1)	n.s.
Obstructive sleep apnea syndrome, n (%)	3 (3.3)	2 (2.9)	1 (4.5)	n.s.
Eosinophilic granulomatosis with polyangiitis, n (%)	10 (11.1)	7 (10.3)	3 (13.6)	n.s.
Allergic bronchopulmonary aspergillosis, n (%)	8 (8.9)	5 (7.3)	3 (13.6)	n.s.
Bronchiectasis, n (%)	8 (8.9)	5 (7.3)	3 (13.6)	n.s.
Chronic spontaneous urticaria (CSU), n (%)	6 (6.7)	4 (5.9)	2 (9.1)	n.s.

**Table 2 biomedicines-10-00200-t002:** Comparison between responders and non-responders at T0, T1, T2, T3, and T4.

						**T0**	**T1**	**T2**
**Variables** **Patient Number**						**R** **68**	** *p* **	** *p* **
Exacerbations rate, n/year						2.3 ± 1.2	<0.001	<0.001
ACT						18.8 ± 3.1	0.021	0.006
FEV1 (% of predicted value)						78.8 ± 19.7	0.048	0.005
FEV1/VC ratio (%)						68.3 ± 11.4	n.s.	0.034
F_E_NO (ppb)						46.5 ± 30.9	n.s.	n.s.
Patients on chronic OCS, n (%)						28 (41.2)	0.039	0.048
	**T3**				**T4**			
**Variables**	**Overall**	**R**	**NR**	** *p* **	**Overall**	**R**	**NR**	** *p* **
**Patient Number**	**53**	**36**	**17**		**39**	**25**	**14**	
Exacerbations rate, n/year	0.4 ± 0.7	0.3 ± 0.6	0.8 ± 0.7	0.023	0.6 ± 0.7	0.5 ± 0.6	0.7 ± 0.9	n.s.
ACT	22.7 ± 3.4	23.5 ± 1.8	20.9 ± 5.4	0.008	22.3 ± 3.4	23.2 ± 1.7	20.7 ± 5.0	0.028
FEV1 (% of predicted value)	88.7 ± 19.1	90.8 ± 19.5	83.6 ± 17.7	n.s.	82.4 ± 19.2	85.9 ± 18.4	76.4 ± 19.7	n.s.
FEV1/VC% ratio (%)	66.7 ± 11.1	68.1 ± 12.7	70.1 ± 9.3	n.s.	65.4 ± 12.4	66.4 ± 10.2	65.7 ± 17.3	n.s.
F_E_NO (ppb)	26.4 ± 21.6	23.2 ± 18.1	34.1 ± 27.3	0.049	19.9 ± 11.2	19.1 ± 10.5	21.3 ± 12.6	n.s.
Patients on chronic OCS, n (%)	16 (30.2)	9 (25)	7 (41)	0.048	9 (23)	3 (12)	6 (43)	0.007

**Table 3 biomedicines-10-00200-t003:** Trends in the exacerbations rate (n/year) and ACT score over time in responders and non-responders; comparisons with baseline.

(a) Exacerbations Rate (n/year)						(b) ACT					
R			NR			R			NR		
T0 (68 patients)	2.3 ± 1.2		T0 (22 patients)	3.1 ± 1.3		T0 (68 patients)	18.8 ± 3.1		T0 (22 patients)	17.4 ± 3.9	
T1 (68 patients)	0.6 ± 0.8		T1 (22 patients)	2.0 ± 1.0		T1 (68 patients)	22.7 ± 2.6		T1 (22 patients)	20.4 ± 4.4	
T2 (55 patients)	0.3 ± 0.6		T2 (18 patients)	1.8 ± 0.7		T2 (55 patients)	22.9 ± 2.5		T2 (18 patients)	20.5 ± 4.7	
T3 (36 patients)	0.3 ± 0.6		T3 (17 patients)	0.8 ± 0.7		T3 (36 patients)	23.5 ± 1.8		T3 (17 patients)	20.9 ± 5.4	
T4 (25 patients)	0.5 ± 0.6		T4 (14 patients)	0.7 ± 0.9		T4 (25 patients)	23.2 ± 1.7		T4 (14 patients)	20.7 ± 5.0	
	*p*			*p*			*p*			*p*	
T0	T1	<0.001	T0	T1	n.s.	T0	T1	<0.001	T0	T1	n.s.
	T2	<0.001		T2	n.s.		T2	<0.001		T2	n.s.
	T3	<0.001		T3	0.001		T3	<0.001		T3	0.001
	T4	<0.001		T4	0.001		T4	<0.001		T4	0.001

**Table 4 biomedicines-10-00200-t004:** Risk factors for switching biological drugs.

Variable	Odds Ratio (95%CI)	χ^2^
Exacerbations (3 or more)	3.75 (1.22–11.41)	0.022
ACT score (less than 17)	2.961 (1.00–8.74)	0.047
Past smoking	7.17 (1.16–43.97)	0.018
Nasal polyps	2.59 (0.85–7.84)	0.047

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
