# Peer review of "Biologics in Severe Eosinophilic Asthma: Three-Year Follow-Up in a SANI Single Center"

_biomedicines, 2022, doi:10.3390/biomedicines10020200_

Round 1
Reviewer 1 Report
The manuscript looks much better after extensive revision , describes a consistent work behind and brings a relevant contribution in the field
One more suggestion : there are too many key words , usually 5-10 are recommended . I would keep : severe eosinophilic asthma, biologics , Th2-high asthma, SANI registry, real life setting
Reviewer 2 Report
The authors have revised all the suggested changes positively in the revised version.
This manuscript is a resubmission of an earlier submission. The following is a list of the peer review reports and author responses from that submission.
Round 1
Reviewer 1 Report
- It’s hard to realize the criteria of exacerbation in the article because the definition is unclear.
- The hospitalization means all-cause hospitalization, respiratory hospitalization, or asthma-related hospitalization? The definition of hospitalization is unclear.
- The definition of severe asthma is inaccurate. The clinical criteria of severe asthma need to clarify in the manuscript. [Please author should summarize the definition of severe asthma in this study according to the GINA guidelines, in order to facilitate reader to follow.]
- The study design is unclear in this article. It is retrospective design or prospective design?
- Authors mention that “due to the COVID-19 outbreak, lung function tests and FeNO measurements had not been performed at the time of visits, thus reducing the data available for statistical analysis”. However, authors did not mention that the method for dealing with missing value of lung function tests and FeNO measurements.
- Author identified that risk factors for biologic drug switching were concomitant COPD, chronic treatment with OCS (p<0.05) and lower baseline FeNO level (p=0.022) which represent these characteristics of patients associated with a r risk of switching therapy. However, authors only compare the characteristics of the patient received biologic drug who were switched and who were not. The methods seem inappropriate for risk factor analysis.
- The discussion is poor and mixed result which is hard to read and understand the concept.
- Although widely accepted threshold indicative of a meaningful improvement was not available, the previous study defines the thresholds considered were an improvement of 80 mL or more and 10% or more from baseline in prebronchodilator FEV1 has been used. In presents article, author use prebronchodilator or post-bronchodilator for lung function test? Could authors define the thresholds considered were an improvement of lung function to provide a meaningful improvement?
Statistical method - Page 3 Line 101: this three statistical methods was used to normality test, which one is carried out in this study?
- Page 3 Line 104: Are all of continuous variables presented as mean +- SD even if author carried out a normality test before descriptive analysis?
- All results are based on repeated measures ANOVA analysis. This is not a reliable result for determining the trend of biologics. It may be that the value of biologics will be affected by the patient's age, gender, and medication used or etc. Therefore, we suggest that the author should perform a multivariate analysis, such as GEE or ANCOVA analysis, to adjust for potential confounding factors.
Reviewer 2 Report
Congratulations on the work; although this is a study with a long follow-up in severe asthma, I believe that the study has some gaps that should be improved.
The mode of inclusion of patients is not described in detail, even if it is severe asthma. This is particularly important in an open-label study, as the investigator's discretion to include or not to include a subject may bias the results. Were all patients who attended the consultations included, which ones were excluded and for what reasons?
There is a high percentage of patients who do not respond to the first biologic, 25%, and yet it is not analysed to what extent this could be due, perhaps the drug chosen was not the right one? Patient characteristics? I believe that this and other observed results should be discussed in more detail.
Reviewer 3 Report
- Interesting and actual topic , useful for clinical practice
- Key words and conclusions are missing
- English has to be checked and improved - examples :
- Line 61- diagnosed with uncontroled...
- Line 68 – signed informed consent
- Line 217- rephrasing needed
- Data refering to patients compliance to therapy and inhalation technique before initiation of biologicals might be added
- Suggested references :
- Leru PM, Anton VF. Real-life benefit of Omalizumab in improving control of bronchial asthma during COVID-19 pandemic. Case report. Cureus , 2021
- Leru PM. Biomarkers in asthma-interpretation and utily in current asthma management. Review. Current Respiratory Medicine Reviews, Special Issue, 2020